# DANA: Scalable Out-of-the-box Distributed ASGD Without Retuning

## Abstract

Distributed computing can significantly reduce the training time of neural networks. Despite its potential, however, distributed training has not been widely adopted: scaling the training process is difficult, and existing SGD methods require substantial tuning of hyperparameters and learning schedules to achieve sufficient accuracy when increasing the number of workers. In practice, such tuning can be prohibitively expensive given the huge number of potential hyperparameter configurations and the effort required to test each one.

We propose DANA, a novel approach that scales out-of-the-box to large clusters using the same hyperparameters and learning schedule optimized for training on a single worker, while maintaining similar final accuracy without additional overhead. DANA estimates the future value of model parameters by adapting Nesterov Accelerated Gradient to a distributed setting, and so mitigates the effect of gradient staleness, one of the main difficulties in scaling SGD to more workers.

Evaluation on three state-of-the-art network architectures and three datasets shows that DANA scales as well as or better than existing work without having to tune any hyperparameters or tweak the learning schedule. For example, DANA achieves 75.73% accuracy on ImageNet when training ResNet-50 with 16 workers, similar to the non-distributed baseline.

## 1 Introduction

Modern deep neural networks are comprised of millions of parameters, which require massive amounts of data and time to learn. Steady growth of these networks over the years has made it impractical to train them from scratch on a single GPU. Distributing the computations over several GPUs can drastically reduce this training time. Unfortunately, stochastic gradient descent (SGD), typically used to train these networks, is an inherently sequential algorithm. As a result, training deep neural networks on multiple *workers* (computational devices) is difficult, especially when trying to maintain high efficiency, scalability and final accuracy.

*Data Parallelism* is a common practice for distributing computation: data is split across multiple workers and each worker computes over its own data. *Synchronous SGD* (SSGD) is the most straightforward method to distribute the training process of neural networks: each worker computes the gradients over its own separate mini-batches, which are then aggregated to update a single model. The result is identical to multiplying the batch size $B$ by the number of workers $N$, so the *effective batch size* is $B \cdot N$. This severely limits scalability and reduces the model accuracy if not handled carefully (Smith et al., 2018; Devarakonda et al., 2017; Goyal et al., 2017). Furthermore, synchronization limits SSGD progress to the slowest worker: all workers must finish their current mini-batch and update the parameter server before any can proceed to the next mini-batch.

*Asynchronous SGD* (ASGD) addresses these drawbacks by removing synchronization between the workers (Dean et al., 2012). Unfortunately, it suffers from *gradient staleness*: gradients sent by workers are often based on parameters that are older than the *master's* (parameter server) current parameters. Hence, distributed ASGD suffers from slow convergence and reduced final accuracy, and may not converge at all if the number of workers is high (Zhang et al., 2016b). Several works attempt to address these issues (Zheng et al., 2017; Zhang et al., 2015; 2016b; Dean et al., 2012), but none has managed to overcome these problems when scaling to a large number of workers.

More crucially, many ASGD algorithms require re-tuning of hyperparameters when scaling to different numbers of workers, and several even introduce new hyperparameters that must also be tuned (Zheng et al., 2017; Zhang et al., 2015; 2016b). In practice, the vast number of potential hyperparameter configurations means that tuning is often done in parallel, with each worker independently evaluating a single configuration using standard SGD. Once the optimal hyperparameters are selected, training is completed on larger clusters of workers. Any additional tuning for ASGD can thus be computationally expensive and time-consuming. Though many algorithms have been proposed to reduce the cost of tuning (Bergstra & Bengio, 2012; Li et al., 2017; Klein et al., 2017; Hazan et al., 2018; Snoek et al., 2015), hyperparameter search remains a significant obstacle, and many practitioners cannot afford to re-tune hyperparameters for distributed training.

**Our contribution:** We propose *Distributed Accelerated Nesterov ASGD* (DANA), a new distributed ASGD algorithm that works *out of the box*: it achieves state-of-the-art accuracy on existing architectures without any additional hyperparameter tuning or changes to the training schedule, while scaling as well or better than existing ASGD approaches, and without any additional overhead. Our DANA implementation achieves state-of-the-art accuracy on ImageNet when training ResNet-50 with 16 and even 32 workers, as well as on CIFAR-10 and CIFAR-100.

Table 1: DANA scales out-of-the-box (OOTB) as well as state of the art asynchronous methods without compromising baseline accuracy. See Section 5 and Section 6 for details.

| Dataset | $N$ | Algorithm | OOTB | Network | Additional Error[a] |
|---|---|---|---|---|---|
| ImageNet | 16 | DC-ASGD (Zheng et al., 2017) | No | ResNet-50 | $+0.48\%$ |
| | 16 | AD-PSGD (Lian et al., 2018) | No | ResNet-50 | $+0.02\%$ |
| | 16 | DANA | **Yes** | ResNet-50 | $-0.35\%$ |
| ImageNet | 32 | AD-PSGD | No | ResNet-50 | $+0.64\%$ |
| | 32 | DANA | **Yes** | ResNet-50 | $+0.65\%$ |
| CIFAR-10 | 8 | DC-ASGD | No | ResNet-20 | $-0.18\%$ |
| | 8 | DANA | **Yes** | ResNet-20 | $-0.24\%$ |
| CIFAR-10 | 16 | EAMSGD (Zhang et al., 2015) | No | ResNet-20 | $+1.43\%$[b] |
| | 16 | AD-PSGD | No | ResNet-20 | $-0.24\%$ |
| | 16 | DANA | **Yes** | ResNet-20 | $-0.08\%$ |

[a]Difference in test error compared to the baseline centralized version.
[b]From Lian et al. (2017).

## 2 BACKGROUND

The goal of SGD is to minimize an optimization problem $J(\theta)$ where $J$ is the objective function (i.e., loss) and the vector $\theta \in R^k$ is the model's parameters. Let $\nabla J$ be the gradient of $J$ with respect to its argument $\theta$. Then the update rule of SGD for the given problem with learning rate $\eta$ is:

$$g_t = \nabla_\theta J(\theta_t)$$
$$\theta_{t+1} = \theta_t - \eta g_t \tag{1}$$

**Momentum** Momentum (Polyak, 1964) has been demonstrated to accelerate SGD convergence and reduce oscillation (Sutskever et al., 2013). Momentum can be compared to a heavy ball rolling downhill that accumulates speed on its way towards the minima. Mathematically, the momentum update rule is obtained by adding a fraction $\gamma$ of the previous update vector $v_{t-1}$ to the current update vector $v_t$:

$$g_t = \nabla J(\theta_t)$$
$$v_{t+1} = \gamma v_t + g_t \tag{2}$$
$$\theta_{t+1} = \theta_t - \eta v_{t+1}$$

When successive gradients have similar direction, momentum results in larger update steps (higher speed), yielding up to quadratic speedup in convergence rate for stochastic and standard gradient descent (Loizou & Richtárik, 2017b;a).

---

**Algorithm 1** ASGD: worker

Receive parameters $\theta_t$ from master
Compute gradients $g_t \leftarrow \nabla J(\theta_t)$
Send $g_t$ to master

---

**Algorithm 2** ASGD: master

Receive gradients $g_t$ from worker $i$ (at iteration $t + \tau$)
$\theta_{t+\tau+1} \leftarrow \theta_{t+\tau} - \eta g_t$
Send parameters $\theta_{t+\tau+1}$ to worker $i$

---

**Nesterov**   Continuing the analogy of a heavy ball rolling downhill, higher speed might make the heavy ball overshoot the bottom of the valley (the local or global minima) if it does not slow down in time. Nesterov (1983) proposed *Nesterov Accelerated Gradient* (NAG), which gives the ball a "sense" of where it is going, allowing it to slow down in advance. Formally, NAG approximates $\hat{\theta}_t$, the future value of $\theta_t$, using the previous update vector $v_t$: $\hat{\theta}_t = \theta_t - \eta\gamma v_t$, and computes the gradients on the parameters' approximated future value $\hat{\theta}$ instead of their current value $\theta$. This allows NAG to slow down near the minima before overshooting the goal and climbing back up the hill. We call this *look-ahead* since it allows us a peek at $\theta$'s future position. The NAG update rule is identical to Equation 2, except that the gradient $g_t$ is computed on the approximated future parameters $\hat{\theta}_t$ instead of $\theta_t$: $g_t = \nabla J(\hat{\theta}_t)$. It is then applied to the original parameters $\theta_t$ via $v_t$ as in Equation 2.

Equation 3 shows that the difference between the updated parameters $\theta_{t+1}$ and the approximated future position $\hat{\theta}_t$ is only affected by the newly computed gradients $g_t$, and not by $v_t$. Hence, NAG can accurately estimate future gradients even when the update vector $v_t$ is large.

$$\theta_{t+1} - \hat{\theta}_t = \theta_t - \eta v_{t+1} - \theta_t + \eta\gamma v_t$$
$$= \eta\gamma v_t - \eta(\gamma v_t + g_t) = -\eta g_t \tag{3}$$

## 3  GRADIENT STALENESS AND MOMENTUM

In ASGD training, each worker $i$ pulls up-to-date parameters $\theta_t$ from the master and computes gradients of a single batch (Algorithm 1). Once computation has finished, worker $i$ sends the computed gradient $g_t$ back to the master. The master (Algorithm 2) then applies the gradient $g_t$ to its current set of parameters $\theta_{t+\tau}$, where $\tau$ is the *lag*: the number of updates the master has received from other workers while worker $i$ was computing $g_t$.

In other words, gradient $g_t$ is *stale*: it was computed from parameters $\theta_t$ but applied to $\theta_{t+\tau}$. This *gradient staleness* is major obstacle to scaling ASGD: the lag $\tau$ increases as the number of workers $N$ grows, decreasing gradient accuracy, and ultimately reducing the accuracy of the trained model.

**From Lag to Gap**   We denote by $\Delta_\theta = \theta_t - \theta_{t+\tau}$ the difference between the master and worker parameters, and define the *gap* as the sum of layer-wise RMSE: $\mathrm{G}(\Delta_\theta) = \sum_{\psi \in \text{layers}} \mathrm{RMSE}(\psi)$, where for each model layer $\psi$ with $m$ parameters, $\mathrm{RMSE}(\psi) = \|\psi\|/\sqrt{m}$. Ideally, there should be no difference between $\theta_t$ and $\theta_{t+\tau}$: when $\Delta_\theta = 0$, gradients are computed on the same parameters they will be applied to. This is the case for sequential and synchronous methods such as SGD and SSGD. In asynchronous methods, however, more workers result in an increased lag $\tau$ and thus a larger gap, as demonstrated by Figure 1(a). A larger gap means less accurate gradients, since they are computed on parameters that differ significantly from those they will be applied to. Conversely, a smaller gap means that gradients are likely to be more accurate.

**The Effect of Momentum**   While momentum and Nesterov methods improve SGD convergence and accuracy of trained models, they make scaling to more workers more difficult. As Figure 1(b) shows, adding NAG to ASGD exacerbates gradient staleness, even though the lag $\tau$ is unchanged.

Put differently, NAG and momentum increase the gap $\mathrm{G}(\Delta_\theta)$. Let $x^i$ be the variable $x$ for worker $i$ (for the master, $i = 0$) and $x_t^i$ be the value of that variable at the worker's $t$ iteration. For ASGD without momentum or NAG, $\Delta_\theta$ is the sum of gradients[1], $\Delta_\theta^{\mathrm{ASGD}} = \eta \sum_{i=1}^N g_t^i$, whereas in the case of ASGD with NAG, $\Delta_\theta$ is the sum of update vectors: $\Delta_\theta^{\mathrm{NAG\text{-}ASGD}} = \sum_{i=1}^N v_t^i$. Recall that

---

[1]To simplify analysis, we assume that all workers have equal computation power and are applying their gradients to the master in a round-robin order. These assumptions can be removed without loss of generality by keeping track of worker updates and weighting them accordingly.

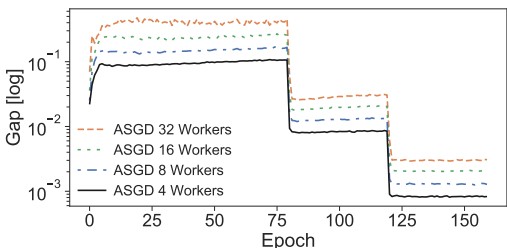 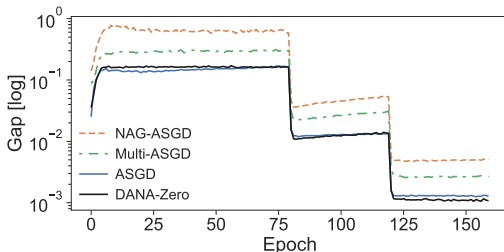

(a) Comparison of the number of workers.  (b) Algorithm comparison (all with 8 workers).

Figure 1: The gap between $\theta_t$ and $\theta_{t+\tau}$ while training ResNet-20 on the CIFAR-10 dataset with (a) different numbers of workers, and (b) different asynchronous algorithms. Adding workers or using momentum increases the effect of the lag $\tau$ on the gap. The large drops in $\mathrm{G}(\theta_t - \theta_{t+\tau})$ are caused by learning rate decay.

by design, momentum and NAG increase the magnitude of updates to $\theta_t$: $\|v_t\| \geq \|g_t\|$. Moreover, if the distribution of training data in each worker is the same (e.g., the common case of assigning data to workers uniformly at random), then the directions of updates $v_t^i$ are approximately similar, since the loss surfaces are similar. Applying the identity $\|a + b\|^2 = \|a\|^2 + \|b\|^2 + 2\langle a, b\rangle$ and the triangle inequality, it follows that in general the gap with NAG is larger than the gap without it: $\mathrm{G}(\Delta_\theta^{\text{NAG-ASGD}}) \geq \mathrm{G}(\Delta_\theta^{\text{ASGD}})$.

Figure 1(b) shows that the gap for ASGD with NAG is substantially larger than for ASGD without it. Conversely, DANA-Zero, detailed in the next section, maintains a low gap throughout training even though it also uses NAG.

# 4 DANA: DISTRIBUTED ACCELERATED NESTEROV ASGD

DANA is a distributed optimizer that converges without hyperparameter tuning even when training with momentum on large clusters. It reduces the gap $\mathrm{G}(\Delta_\theta)$ by computing the worker's gradients on parameters that more closely resemble the master's future position $\theta_{t+\tau}$. We extend NAG to the common distributed setting with $N$ workers and one master, obtaining similar look-ahead to the traditional method with a single worker. This means that for the same lag $\tau$, DANA suffers from a reduced gap and therefore suffers less from gradient staleness.

## 4.1 THE DANA-ZERO UPDATE RULE

In DANA-Zero, the master maintains a separate update vector $v^i$ for each worker, which is updated with the worker's gradients $g^i$ using the same update rule as in classic SGD with momentum (Equation 2). Since the master updates each $v^i$ only with the gradients from worker $i$, we can apply look-ahead using the most recent update vectors of the other workers. We know that $v_{t-1}^i$ will move the master's parameters $\theta^0$ on iteration $t$ of worker $i$ by $\eta\gamma v_{t-1}^i$. Thus, computing $\theta_t^0 - \eta\gamma v_{t-1}^i$ gives us an approximation of the next position of the master's parameters after worker $i$ has sent its gradients. Instead of sending the master's current parameters $\theta^0$ to the worker, DANA-Zero sends the estimated future position of the master's parameters after $N$ updates, one for each worker:

$$\hat{\theta}_{\text{DANA}} \triangleq \theta^0 - \eta\gamma \sum_{i=1}^{N} v_{prev(i)}^i \tag{4}$$

where $prev(i)$ denotes the last iteration where worker $i$ sent gradients to the master. Algorithm 3 shows the DANA-Zero master algorithm; the worker code is the same as in ASGD (Algorithm 1).

Given the update rule, we calculate the gap of DANA-Zero, $\mathrm{G}(\Delta_\theta^{\text{DANA}})$, similarly to Equation 3:

$$\Delta_\theta^{\text{DANA}} = \eta \sum_{i=1}^{N} v_t^i - \eta\gamma \sum_{i=1}^{N} v_{t-1}^i = \eta \sum_{i=1}^{N}(v_{t+1}^i - \gamma v_t^i) = \eta \sum_{i=1}^{N} g_t^i \tag{5}$$

**Algorithm 3** DANA-Zero master. DANA-Zero uses the standard ASGD worker (Algorithm 1).

---

Receive gradients $g^i$ from worker $i$
Update worker's momentum $v^i \leftarrow \gamma v^i + g^i$
Update master's weights $\theta^0 \leftarrow \theta^0 - \eta v^i$
Send estimate $\hat{\theta} = \theta^0 - \eta\gamma \sum_{j=1}^{N} v^j$ to worker $i$

**Algorithm 4** DANA worker $i$. DANA uses the standard ASGD master (Algorithm 2).

---

Receive parameters $\Theta^i$ from master
Compute gradients $g^i \leftarrow \nabla J(\Theta^i)$
Update momentum $v^i \leftarrow \gamma v^i + g^i$
Send update step $\gamma v^i + g^i$ to master

Equation 5 shows that DANA-Zero has the same gap as ASGD without momentum. Figure 1(b) demonstrates this empirically: ASGD with momentum has a larger gap than ASGD throughout the training process, whereas DANA-Zero's gap is similar to ASGD despite also using momentum. Additionally, when running with one worker ($N = 1$), DANA-Zero reduces to a single standard NAG optimizer: with one worker, $\theta_t^1 = \theta_t^0 - \eta\gamma$, so merging the master and the worker algorithms yields the Nesterov update rule (see Appendix A for more details).

## 4.2 Optimizing DANA

In DANA-Zero, the master maintains an update vector for every worker, and must also compute $\hat{\theta}$ at each iteration. This adds a computation and memory overhead to the master. DANA is a variation of DANA-Zero that obtains the same look-ahead as DANA-Zero but without any additional memory or computation overhead.

**Bengio-Nesterov Momentum** Bengio et al. (2013) proposed a simplified Nesterov update rule, known as Bengio-Nesterov Momentum. This variation of the classic Nesterov is occasionally used in deep learning frameworks (Paszke et al., 2017) since it simplifies the implementation. Bengio-Nesterov Momentum works by defining a new variable $\Theta$ to represent $\theta$ after the momentum update:

$$\Theta_t \triangleq \theta_t - \eta\gamma v_t \tag{6}$$

Substituting $\theta_t$ with $\Theta_t$ in the NAG update rule (Section 2) yields the Bengio-Nesterov update rule:

$$\theta_{t+1} = \theta_t - \eta v_{t+1} \implies \Theta_{t+1} + \eta\gamma v_{t+1} = \Theta_t + \eta\gamma v_t - \eta v_{t+1}$$
$$= \Theta_t + \eta\gamma v_t - \eta\left(\gamma v_t + \nabla J(\Theta_t)\right)$$
$$\implies \Theta_{t+1} = \Theta_t - \eta(\gamma v_{t+1} + \nabla J(\Theta_t)) \tag{7}$$

Using Equation 7, an implementation need only store one set of parameters in memory ($\Theta$) since gradients are both computed from and applied to $\Theta$, rather than computed on $\hat{\theta}$ but applied to $\theta$.

**The DANA Update Rule** We leverage the ideas of Bengio-Nesterov Momentum to optimize DANA. As we did in Equation 6, we define a new variable $\Theta$ that represents $\theta$ after the momentum update from all workers:

$$\Theta_t \triangleq \theta_t - \eta\gamma \sum_{j=1}^{N} v_{prev(j)}^j \tag{8}$$

We define $\Theta_{t+1}$ as $\Theta_t$ after applying worker $i$'s update $v_{t+1} = \gamma v_t + \nabla J(\Theta_t)$:

$$\Theta_{t+1} = \theta_{t+1} - \eta\gamma\left(v_{prev(i)+1}^i + \sum_{j\neq i} v_{prev(j)}^j\right)$$

Substituting $\theta_t$ with $\Theta_t$ yields the DANA update rule (Equation 9):

$$\theta_{t+1} = \theta_t - \eta v_{prev(i)+1}^i$$

$$\implies \Theta_{t+1} + \eta\gamma\left(v_{prev(i)+1}^i + \sum_{j\neq i} v_{prev(j)}^j\right) = \Theta_t + \eta\gamma \sum_{j=1}^{N} v_{prev(j)}^j - \eta v_{prev(i)+1}^i$$

$$\implies \Theta_{t+1} = \Theta_t + \eta\gamma(v_{prev(i)}^i - v_{prev(i)+1}^i) - \eta v_{prev(i)+1}^i$$

$$= \Theta_t - \eta(\gamma v_{prev(i)+1}^i + \nabla J(\Theta_t)) \tag{9}$$

Algorithm 4 shows DANA: a variation of DANA-Zero that uses Bengio-Nesterov to eliminate the overhead at the master. DANA only changes the worker side and uses the same master algorithm as in ASGD (Algorithm 2); hence, it eliminates any additional overhead at the master. DANA is equivalent to DANA-Zero in all other ways, and provides the same benefits: it works out-of-the-box, provides look-ahead to reduce the gap[2] and achieves the same fast convergence and high accuracy.

## 5 EVALUATION

We implemented DANA using PyTorch (Paszke et al., 2017) and mpi4py (Dalcin et al., 2011) and evaluated it by: (a) simulating multiple distributed workers[3] on a single machine to focus on accuracy rather than communication overheads and update scheduling; and (b) running the distributed algorithm on multiple machines, where we measure run time speedups and confirm simulation accuracy. We simulate two modes. In *block-random scheduling* every block of $N$ updates contains one update from each worker and order is shuffled between blocks, which simulates the common case where distributed workers have very similar computational power. In the *gamma-distributed model* the execution time for each individual batch is drawn from a gamma distribution (Ali et al., 2000). The gamma distribution is a well-accepted model for task execution time, and gives rise to stragglers naturally. We use the formulation proposed by Ali et al. (2000) and set $V = 0.1$ and $\mu = B * V^2$, where $B$ is the chosen batch size, yielding a mean execution time of $B$ simulated time units.

Our main evaluation metric is *final test error*: the error achieved by a trained model after training using the baseline training schedule. We also measure improvement in training time (*speedup*) using the distributed DANA implementation.

**Algorithms**    As we are interested in out-of-the-box performance, we compare DANA to algorithms that do not introduce new parameters and require no re-tuning (see Table 1 for comparison to non-OOTB methods). All runs use the same hyperparameters, training schedule and data augmentation from the original paper where the network architectures are proposed.

1. *Baseline:* single worker with the same hyperparameters as in the respective NN paper.
2. *SSGD*: similar to Goyal et al. (2017) with the linear scaling rule.
3. *ASGD:* standard asynchronous SGD without momentum (momentum parameter set to 0).
4. *NAG-ASGD:* asynchronous SGD which uses a single NAG optimizer for all workers.
5. *Multi-ASGD:* asynchronous SGD which holds a separate NAG optimizer for each worker.
6. *DANA:* DANA as described in Section 4.2.

In the early stages of training, the network changes rapidly, which can lead to training error spikes. For all algorithms, we follow the gradual warm-up approach proposed by Goyal et al. (2017) to overcome this problem: we divide the initial training rate by the number of workers $N$ and ramp it up linearly until it reaches its original value after 5 epochs. We also use momentum correction (Goyal et al., 2017) in all algorithms to stabilize training when the learning rate changes.

**Datasets**    We evaluated DANA on CIFAR-10, CIFAR-100 (Hinton, 2007) and ImageNet (Russakovsky et al., 2015). The CIFAR-10 Hinton (2007) dataset is comprised of 60k RGB images partitioned into 50k training images and 10k test images. Each image contains 32x32 RGB pixels and belongs to one of ten equal-sized classes. CIFAR-100 is similar but has 100 classes. The ImageNet dataset (Russakovsky et al., 2015), known as ILSVRC2012, consists of RGB images, each labeled as one of 1000 classes. Images are partitioned to 1.28 million training images and 50k validation images, and each image is randomly cropped and re-sized to 224x224 (1-crop validation).

### 5.1 OUT-OF-THE-BOX ACCURACY

Figure 2 shows the mean and standard deviation of final test error from five runs using block-random scheduling, when training the ResNet-20 (He et al., 2016) architecture on CIFAR-10 and the Wide ResNet 16-4 (Zagoruyko & Komodakis, 2016) architecture on CIFAR-10 and CIFAR-100.

---

[2]DANA uses $\Theta$ internally, so computing the gap $G(\Delta_\theta)$ would require transforming all values back to $\theta$.

[3]A single worker may not be a single GPU. DANA, like all ASGD algorithms, can treat each machine with multiple GPUs as a single worker. For example, DANA can run on 32 workers with 8 GPUs each, where each worker performs SSGD internally (which is transparent to the ASGD algorithm).

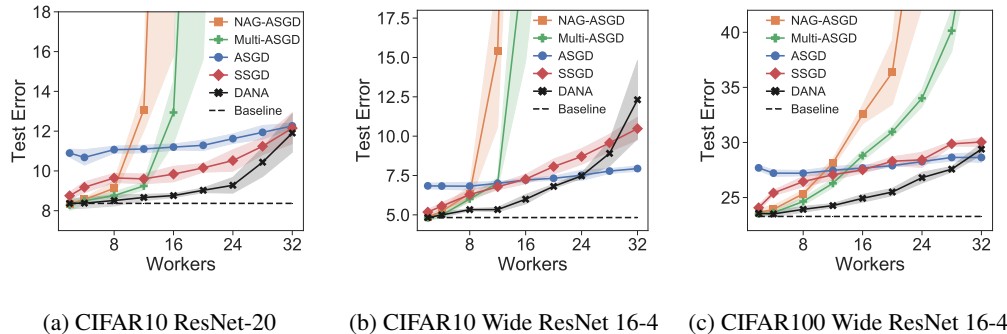

(a) CIFAR10 ResNet-20     (b) CIFAR10 Wide ResNet 16-4     (c) CIFAR100 Wide ResNet 16-4

Figure 2: Final test error for different numbers of workers $N$ on the CIFAR10 and CIFAR100 datasets using ResNet-20 and Wide ResNet 16-4 using block-random scheduling. Bold lines show the mean over the 5 different experiments, while transparent bands show the standard deviation. The baseline is the mean of 5 different runs with a single worker.

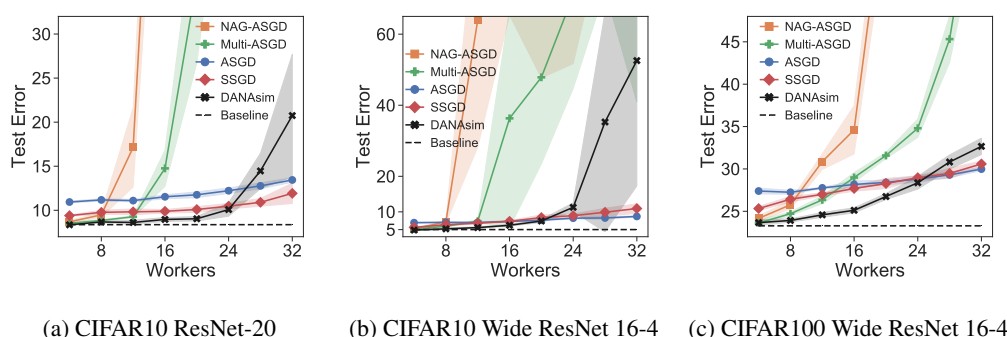

(a) CIFAR10 ResNet-20     (b) CIFAR10 Wide ResNet 16-4     (c) CIFAR100 Wide ResNet 16-4

Figure 3: Final test error for different numbers of workers $N$ on the CIFAR10 and CIFAR100 datasets using ResNet-20 and Wide ResNet 16-4 using the gamma-distributed model of execution time. Bold lines show the mean over the 5 different experiments, while transparent bands show the standard deviation. The baseline is the mean of 5 different runs with a single worker.

Out-of-the-box, DANA's final test error remains similar to the baseline error with up to 24 workers in Figure 2(a) and 12 workers in Figures 2(b) and 2(c). Moreover, DANA's final error is lower than the other algorithms when using up to 24–32 workers – all without any tuning. Above that point, DANA is no longer the superior algorithm because of the smaller size of CIFAR-10 and CIFAR-100: with so many workers the amount of data per worker is so small that gradients from different workers become dissimilar, and DANA is no longer able to mitigate the effects of momentum. ImageNet results (Table 2) show that DANA easily scales to 32 workers when there is enough data per worker.

NAG-ASGD demonstrates the detrimental effect of momentum on gradient staleness: it yields good accuracy with few workers, but test error climbs sharply and sometimes even fails to converge when used with more than 16 workers. On the other hand, even though ASGD without NAG appears to be the most scalable algorithm, its test error is unacceptably high even with 2 workers. While SSGD appears to offer a middle ground of reasonable accuracy with good scalability, in practice speedup is limited by synchronization and the increase in effective batch size means tuning is required to achieve good accuracy. DANA provides a way out of this dilemma: by mitigating gradient staleness, it achieves the best final accuracy while scaling to many workers, and works without changing any hyperparameter or changing the learning schedule.

Finally, Multi-ASGD serves as an ablation study: its poor scalability demonstrates that it is not sufficient to simply maintain update vectors for every worker. The DANA update rules (Section 4) are also required to achieve a high test accuracy.

Table 2: Out-of-the-box ResNet-50 ImageNet test errors. Baseline error from He et al. (2016) with block-random scheduling.

| Workers | Algorithm | Test Error |
|---|---|---|
| 1 | Baseline | 24.70 |
| 16 | ASGD | +3.78 |
| | NAG-ASGD | +1.33 |
| | SSGD | +2.17 |
| | **DANA** | **-0.35** |
| 32 | ASGD | +4.59 |
| | NAG-ASGD | +4.57 |
| | SSGD | +3.16 |
| | **DANA** | **+0.65** |
| 48 | DANA | **+1.13** |
| 64 | ASGD | +6.90 |
| | NAG-ASGD | +8.01 |
| | DANA | **+4.10** |

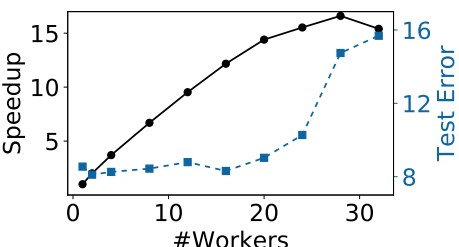

Figure 4: DANA speedup (solid line) and final test error (dashed) when training ResNet-20 on CIFAR-10 with different numbers of workers.

Figure 3 is similar to Figure 2 except that the execution time for each individual batch is drawn from a $\Gamma(100, 1.28)$ gamma distribution (since the batch size $B$ is 128). It shows the mean and standard deviation of final test error from five runs, when training the ResNet-20 (He et al., 2016) architecture on CIFAR-10 and the Wide ResNet 16-4 (Zagoruyko & Komodakis, 2016) architecture on CIFAR-10 and CIFAR-100. The trends in Figure 3 agree with those on Figure 2: up to $N = 24$ workers, DANA performs similar to the baseline and is superior to the other OOTB algorithms.

Table 2 lists out-of-the-box test errors when training the ResNet-50 architecture (He et al., 2016) on ImageNet. Due to the long training time of ImageNet, we only conducted experiments on ImageNet with SSGD, ASGD and DANA. DANA consistently outperforms all other out-of-the-box algorithms. Similar test-errors to Table 2 were achieved when training DANA with the gamma-distributed model on 32 and 64 workers, yielding a final test-error of $+0.54\%$ and $+5.84\%$ respectively. Table 1 compares DANA to reported results from state-of-the-art asynchronous algorithms that rely on tuning or changes to the learning rate schedule, while DANA converges to the ImageNet's baseline test accuracy with 16 and 32 workers, matching or exceeding recent state-of-the-art algorithms (AD-PSGD and DC-ASGD), despite making no changes to any hyperparameter.

## 5.2 SPEEDUP

While this work focuses on improving out-of-the-box ASGD accuracy without adding overhead, we also measured speedup, defined as the runtime for DANA with $N$ workers divided by the runtime for the single worker baseline. Figure 4 shows the speedup and final test error when running DANA on the Google Cloud Platform with a single parameter server (master) and one Nvidia Tesla V100 GPU per machine, when training ResNet-20 on the CIFAR-10 dataset. It shows speedup of up to $\times 16$ when training with $N = 24$ workers, and as before, its final test error remains close to the baseline up to $N = 24$ workers.

At 24 workers, the parameter server becomes a bottleneck. This phenomenon is consistent with literature (Xing et al., 2015) on ASGD, and is well-studied. Since the DANA master is unchanged from the ASGD algorithm (Algorithm 2), existing techniques, such as sharding the parameter server (Dean et al., 2012), improving network utilization (Li et al., 2014), lock-free synchronizations (Recht et al., 2011; Zhang et al., 2016a), and gradient compression (Lin et al., 2018; Wen et al., 2017; Bernstein et al., 2018), are fully compatible with DANA but are beyond the scope of this work.

Figure 5(a) shows the theoretically achievable speedup, based on the detailed gamma-distributed model, for asynchronous (DANA and other ASGD variants) and synchronous algorithms. The asynchronous algorithms can achieve linear speedup;; the synchronous algorithm (SSGD) falls short as

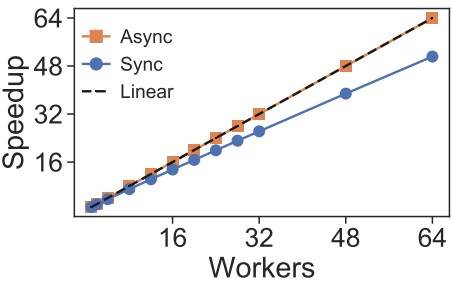 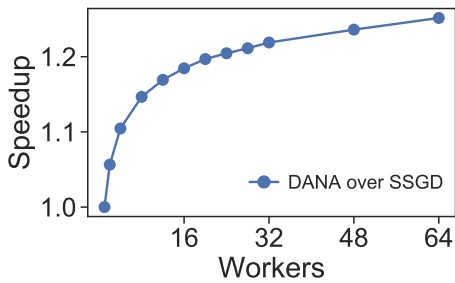

(a) Async (DANA, ASGD) and sync (SSGD) speedup.          (b) DANA speedup over SSGD.

Figure 5: Theoretical speedups for DANA (or any ASGD) and SSGD when batch execution times are drawn from a Gamma distribution, as in Figure 3. Overheads are not modeled.

we increase the number of workers, since it must wait after each iteration until all workers complete their batch. Figure 5(b) shows that DANA (or any ASGD variant) is up to 21% faster than SSGD. This speedup is an underestimate, since our simulation only includes batch execution times, and does not model execution time of barriers, all-gather operations, and other overheads.

## 6    RELATED WORK

DANA achieves out-of-the-box scaling by explicitly mitigating the effects of gradient staleness. Other approaches to mitigating staleness include DC-ASGD (Zheng et al., 2017), which uses a Taylor expansion to approximate the gradients as if they were calculated on the master's recent parameters. DC-ASGD requires substantial tuning of several hyperparameters, introduces additional hyperparameters that must also be tuned, and requires additional computation at the master to approximate the Hessian. Elastic Averaging SGD (EASGD) (Zhang et al., 2015) is an ASGD algorithm that uses a *center force* to pull the workers' parameters towards the master's parameters. This allows each worker to train asynchronously and synchronize with the master once every few batches. However, EASGD introduces three new hyperparameters that must be tuned. Zhang et al. (2016b) proposed Staleness-aware ASGD: worker gradients are weighted by the lag between two successive updates, so stale gradients have lower impact. This method adds one new hyperparameter, and achieves lower or equivalent final accuracy compared to SSGD. DANA scales without adding hyperparameters or tuning, and achieves final accuracy comparable to that of a single worker.

Other approaches to scaling are SSGD learning rate schedulers. Goyal et al. (2017) introduced a linear scaling rule and warmup epochs to help increase the mini-batch size, which is key to scaling the number of workers in a synchronous environment. You et al. (2017) further generalize that work and introduce LARS, a method that changes the learning rate independently for each layer, according to the ratio between norm of the layer's weights and the norm of the layer's current gradient, whose parameters need to be tuned. Smith et al. (2018) suggest increasing the batch size instead of decaying the learning rate. These approaches are compatible with (and indeed orthogonal to) DANA.

Finally, decentralized approaches to scaling SGD eliminate the parameter server entirely. In D-PSGD (Lian et al., 2017), workers first compute and apply gradients locally and then synchronously average models with their neighbors. Very recently, Lian et al. (2018) proposed AD-PSGD, which operates asynchronously. While they demonstrate impressive scaling, these works focus on different communication topologies and use other learning schedules and batch sizes than the baselines.

## 7    DISCUSSION

DANA is a new asynchronous SGD algorithm for training of neural networks. By mitigating the effect of gradient staleness, DANA scales out-of-the-box to large clusters using the same hyperparameters and learning schedule optimized for training on a single worker, while maintaining similar

final accuracy, without adding any overhead at the master. DANA could be used to extend other non-distributed optimization procedures (e.g., Nadam Dozat (2016)) to a distributed setting without adding parameters. Integrating DANA with DC-ASGD could further mitigate gradient staleness, though without eliminating tuning. Finally, we are working to extend DANA with separate, self-adjusting weights per worker to address settings with heterogeneous workers while avoiding tuning.

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

# A  SUPPLEMENTARY MATERIALS

## A.1  DANA-ZERO EQUIVALENCE TO NESTEROV

When running with one worker $(N = 1)$ DANA-Zero reduces to a single NAG optimizer. This can be shown by merging the worker and master (Algorithms 1 and 3) into a single algorithm: since at all times $\theta_t^1 = \theta_t^0 - \eta\gamma v_{t-1}$, the resulting algorithm trains one set of parameters $\theta$, which is exactly the Nesterov update rule. Algorithm 5 shows the fused algorithm, equivalent to standard NAG optimizer.

---

**Algorithm 5** Fused DANA-Zero worker/master (when $N = 1$)

---

Compute gradients $g_t \leftarrow \nabla J(\theta_t - \eta\gamma v_{t-1})$
Update momentum $v_t \leftarrow \gamma v_{t-1} + g_t$
Update weights $\theta_{t+1} \leftarrow \theta_t - \eta v_t$

---

## A.2  EXPERIMENTAL RESULTS ON CIFAR

This section shows the results of the ResNet-20 He et al. (2016) and Wide ResNet 16-4 (16 depth and 4 width) network architectures on the CIFAR-10 and CIFAR-100 datasets. We ran each experiment five times to show the mean and standard deviation of the final test error, which are shown in the tables below. The experiments were executed using two types of worker update orders:

- Round Robin: Every worker updates the master in a sequential order. For example, if $N = 4$, the order of updates is $1, 2, 3, 4, 1, 2, 3, 4 \ldots$.
- Block Random: Every worker updates the master in a random order. However, every $N$ updates, it is guaranteed that every worker has updated the master exactly once.
- Gamma Distribution: Every worker updates the master in a gamma distribution order (Ali et al., 2000). The gamma distribution is a well-accepted model for task execution time, and gives rise to stragglers naturally. We use the formulation proposed by Ali et al. (2000) and set $V = 0.1$ and $\mu = B * V^2$, where $B$ is the chosen batch size. When the batch size is 128 (as in both CIFAR datasets for example) this yields the distribution $\Gamma(100, 1.28)$ with mean execution time of 128 simulated time units.

### A.2.1  RESNET-20 RESULTS

Tables 3 and 4 show the final test error of the ResNet-20 architecture whose training schedule He et al. (2016) starts with an initial learning rate of $0.1$, which decays by a factor of ten on epochs 80 and 120. The batch size is 128, momentum is 0.9, and the baseline uses NAG.

Table 3: ResNet CIFAR10 Test Error Round Robin

| #Workers | Algorithm | mean | std |
|---|---|---|---|
| 1.0 | Baseline | 8.37 | 0.22 |
| 4.0 | ASGD | 11.00 | 0.17 |
|  | DANA | 8.44 | 0.15 |
|  | Multi-ASGD | 8.46 | 0.14 |
|  | NAG-ASGD | 8.50 | 0.27 |
| 8.0 | ASGD | 10.89 | 0.22 |
|  | DANA | 8.55 | 0.22 |
|  | Multi-ASGD | 8.70 | 0.14 |
|  | NAG-ASGD | 9.36 | 0.13 |
| 16.0 | ASGD | 11.79 | 0.24 |
|  | DANA | 8.67 | 0.20 |
|  | Multi-ASGD | 15.05 | 0.53 |
|  | NAG-ASGD | 83.27 | 9.15 |
| 32.0 | ASGD | 13.21 | 0.35 |
|  | DANA | 24.59 | 12.18 |
|  | Multi-ASGD | 83.22 | 11.19 |
|  | NAG-ASGD | 84.98 | 10.00 |

Table 4: ResNet CIFAR10 Test Error Block Random

| #Workers | Algorithm | mean | std |
|---|---|---|---|
| 1.0 | Baseline | 8.37 | 0.22 |
| 2.0 | ASGD | 10.90 | 0.18 |
| | DANA | 8.37 | 0.24 |
| | Multi-ASGD | 8.28 | 0.24 |
| | NAG-ASGD | 8.35 | 0.15 |
| | SSGD | 8.75 | 0.15 |
| 4.0 | ASGD | 10.68 | 0.35 |
| | DANA | 8.38 | 0.23 |
| | Multi-ASGD | 8.50 | 0.16 |
| | NAG-ASGD | 8.56 | 0.07 |
| | SSGD | 9.18 | 0.24 |
| 8.0 | ASGD | 11.07 | 0.16 |
| | DANA | 8.51 | 0.25 |
| | Multi-ASGD | 8.76 | 0.17 |
| | NAG-ASGD | 9.12 | 0.33 |
| | SSGD | 9.65 | 0.21 |
| 12.0 | ASGD | 11.10 | 0.18 |
| | DANA | 8.66 | 0.14 |
| | Multi-ASGD | 9.24 | 0.17 |
| | NAG-ASGD | 13.08 | 0.96 |
| | SSGD | 9.60 | 0.22 |
| 16.0 | ASGD | 11.20 | 0.14 |
| | DANA | 8.76 | 0.09 |
| | Multi-ASGD | 12.94 | 1.40 |
| | NAG-ASGD | 50.50 | 31.05 |
| | SSGD | 9.84 | 0.35 |
| 20.0 | ASGD | 11.29 | 0.24 |
| | DANA | 9.03 | 0.11 |
| | Multi-ASGD | 42.66 | 24.18 |
| | NAG-ASGD | 82.32 | 15.35 |
| | SSGD | 10.15 | 0.19 |
| 24.0 | ASGD | 11.62 | 0.22 |
| | DANA | 9.28 | 0.35 |
| | DANA-Zero | 9.43 | 0.26 |
| | Multi-ASGD | 71.05 | 16.08 |
| | NAG-ASGD | 90.00 | 0.00 |
| | SSGD | 10.52 | 0.49 |
| 28.0 | ASGD | 11.94 | 0.31 |
| | DANA | 10.44 | 0.65 |
| | Multi-ASGD | 85.33 | 8.80 |
| | NAG-ASGD | 72.55 | 22.11 |
| | SSGD | 11.24 | 0.38 |
| 32.0 | ASGD | 12.26 | 0.13 |
| | DANA | 11.90 | 0.86 |
| | Multi-ASGD | 90.00 | 0.00 |
| | NAG-ASGD | 81.49 | 12.87 |
| | SSGD | 12.15 | 0.70 |

### A.2.2    WIDE RESNET 16-4 RESULTS

Tables 5, 7, 6, and 8 show the final test error of the Wide ResNet 16-4 architecture. For both datasets, CIFAR-10 and CIFAR-100, the training schedule Zagoruyko & Komodakis (2016) starts with an initial learning rate of $0.1$, which decays by a factor of five on epochs 60, 120 and 160. The batch size is 128, momentum is 0.9, and the baseline uses NAG.

Table 5: Wide ResNet CIFAR10 Test Error Round Robin

| #Workers | Algorithm | mean | std |
|---|---|---|---|
| 1.0 | Baseline | 4.83 | 0.12 |
| 4.0 | ASGD | 6.81 | 0.12 |
| | DANA | 4.94 | 0.05 |
| | Multi-ASGD | 4.93 | 0.10 |
| | NAG-ASGD | 5.23 | 0.10 |
| 8.0 | ASGD | 6.94 | 0.20 |
| | DANA | 5.29 | 0.14 |
| | Multi-ASGD | 5.93 | 0.30 |
| | NAG-ASGD | 7.27 | 0.25 |
| 16.0 | ASGD | 7.41 | 0.24 |
| | DANA | 6.68 | 0.32 |
| | Multi-ASGD | 40.78 | 24.92 |
| | NAG-ASGD | 84.27 | 11.45 |

Table 6: Wide ResNet CIFAR100 Test Error Round Robin

| #Workers | Algorithm | mean | std |
|---|---|---|---|
| 1.0 | Baseline | 23.28 | 0.30 |
| 4.0 | ASGD | 27.27 | 0.09 |
| | DANA | 23.48 | 0.21 |
| | Multi-ASGD | 23.74 | 0.10 |
| | NAG-ASGD | 24.08 | 0.16 |
| 8.0 | ASGD | 27.39 | 0.28 |
| | DANA | 23.94 | 0.25 |
| | Multi-ASGD | 24.98 | 0.17 |
| | NAG-ASGD | 25.75 | 0.27 |
| 16.0 | ASGD | 27.76 | 0.39 |
| | DANA | 25.13 | 0.32 |
| | Multi-ASGD | 29.45 | 0.41 |
| | NAG-ASGD | 35.02 | 2.36 |

Table 7: Wide ResNet CIFAR10 Test Error Block Random

| #Workers | Algorithm | mean | std |
|---|---|---|---|
| 1.0 | Baseline | 4.83 | 0.12 |
| 2.0 | ASGD | 6.84 | 0.16 |
|  | DANA | 4.82 | 0.13 |
|  | Multi-ASGD | 4.81 | 0.11 |
|  | NAG-ASGD | 4.84 | 0.09 |
|  | SSGD | 5.18 | 0.14 |
| 4.0 | ASGD | 6.83 | 0.18 |
|  | DANA | 5.00 | 0.16 |
|  | Multi-ASGD | 4.98 | 0.17 |
|  | NAG-ASGD | 5.22 | 0.16 |
|  | SSGD | 5.55 | 0.20 |
| 8.0 | ASGD | 6.82 | 0.22 |
|  | DANA | 5.33 | 0.06 |
|  | Multi-ASGD | 6.01 | 0.13 |
|  | NAG-ASGD | 6.36 | 0.48 |
|  | SSGD | 6.29 | 0.26 |
| 12.0 | ASGD | 7.00 | 0.11 |
|  | DANA | 5.33 | 0.12 |
|  | Multi-ASGD | 7.02 | 0.31 |
|  | NAG-ASGD | 15.43 | 4.33 |
|  | SSGD | 6.78 | 0.18 |
| 16.0 | ASGD | 7.21 | 0.09 |
|  | DANA | 5.99 | 0.27 |
|  | Multi-ASGD | 22.36 | 6.80 |
|  | NAG-ASGD | 77.26 | 25.48 |
|  | SSGD | 7.27 | 0.27 |
| 20.0 | ASGD | 7.32 | 0.12 |
|  | DANA | 6.81 | 0.12 |
|  | Multi-ASGD | 65.00 | 30.62 |
|  | NAG-ASGD | 77.18 | 25.64 |
|  | SSGD | 8.07 | 0.43 |
| 24.0 | ASGD | 7.51 | 0.30 |
|  | DANA | 7.48 | 0.15 |
|  | Multi-ASGD | 79.09 | 21.82 |
|  | NAG-ASGD | 90.00 | 0.00 |
|  | SSGD | 8.71 | 0.45 |
| 28.0 | ASGD | 7.78 | 0.14 |
|  | DANA | 8.91 | 0.72 |
|  | Multi-ASGD | 72.06 | 22.65 |
|  | NAG-ASGD | 90.00 | 0.00 |
|  | SSGD | 9.57 | 0.49 |
| 32.0 | ASGD | 7.94 | 0.15 |
|  | DANA | 12.31 | 2.22 |
|  | Multi-ASGD | 66.71 | 23.14 |
|  | NAG-ASGD | 90.00 | 0.00 |
|  | SSGD | 10.48 | 0.64 |

Table 8: Wide ResNet CIFAR100 Test Error Block-Random

| #Workers | Algorithm | mean | std |
|---|---|---|---|
| 1.0 | Baseline | 23.28 | 0.30 |
| 2.0 | ASGD | 27.69 | 0.22 |
| | DANA | 23.56 | 0.23 |
| | Multi-ASGD | 23.51 | 0.11 |
| | NAG-ASGD | 23.62 | 0.35 |
| | SSGD | 24.08 | 0.23 |
| 4.0 | ASGD | 27.22 | 0.26 |
| | DANA | 23.51 | 0.25 |
| | Multi-ASGD | 23.62 | 0.22 |
| | NAG-ASGD | 23.93 | 0.31 |
| | SSGD | 25.41 | 0.22 |
| 8.0 | ASGD | 27.21 | 0.22 |
| | DANA | 23.93 | 0.24 |
| | Multi-ASGD | 24.64 | 0.31 |
| | NAG-ASGD | 25.34 | 0.30 |
| | SSGD | 26.45 | 0.42 |
| 12.0 | ASGD | 27.47 | 0.32 |
| | DANA | 24.27 | 0.17 |
| | Multi-ASGD | 26.29 | 0.35 |
| | NAG-ASGD | 28.14 | 0.34 |
| | SSGD | 27.07 | 0.42 |
| 16.0 | ASGD | 27.68 | 0.28 |
| | DANA | 24.93 | 0.32 |
| | Multi-ASGD | 28.81 | 0.44 |
| | NAG-ASGD | 32.55 | 0.77 |
| | SSGD | 27.51 | 0.26 |
| 20.0 | ASGD | 27.89 | 0.10 |
| | DANA | 25.50 | 0.30 |
| | Multi-ASGD | 30.97 | 0.42 |
| | NAG-ASGD | 36.36 | 2.61 |
| | SSGD | 28.29 | 0.36 |
| 24.0 | ASGD | 28.26 | 0.22 |
| | DANA | 26.80 | 0.44 |
| | Multi-ASGD | 34.01 | 0.70 |
| | NAG-ASGD | 54.64 | 11.07 |
| | SSGD | 28.42 | 0.62 |
| 28.0 | ASGD | 28.64 | 0.30 |
| | DANA | 27.57 | 0.23 |
| | Multi-ASGD | 40.15 | 1.71 |
| | NAG-ASGD | 80.63 | 7.30 |
| | SSGD | 29.87 | 0.19 |
| 32.0 | ASGD | 28.63 | 0.40 |
| | DANA | 29.39 | 0.47 |
| | Multi-ASGD | 56.10 | 4.15 |
| | NAG-ASGD | 93.26 | 3.82 |
| | SSGD | 30.04 | 0.36 |

