# OpenReview forum: "DANA: Scalable Out-of-the-box Distributed ASGD Without Retuning"
_ICLR.cc/2019/Conference_

### Official Review · AnonReviewer3 · 2018-11-02
**Interesting idea, but evaluation constraints are limiting**

**Rating:** 5
**Confidence:** 4

**Review:**

# overview
In this work, Nesterov Accelerated gradient based updates are applied in a distributed fashion to scale SGD based training to multiple nodes without the introduction of further hyperparameters or having to adapt the learning rate schedule from that of single node training on the same data.

Evaluation is carried out on image classification workloads using ResNet model variants across CIFAR10,100, and ImageNet datasets, utilizing from 8-32 nodes. In contrasting test error relative to single node performance, the authors find their method degrades less than other synchronous and asynchronous SGD based approaches as node count increases.

Overall, this work is presented in a fairly clear and logical manner, and the writing is easy to follow.  However the approach described appears to be contingent on very specific worker communication patterns and timing which seem unrealistic for real-world settings (namely that each worker sends exactly one update per N sized block received).  Extrapolating from the curvature of the results shown it doesn't appear that DANA would continue to outperform other methods like ASGD once the worker count scales beyond the 32 node limit evaluated.

# pros
* no additional hyperparameter tuning required
* should be easy to drop into existing asynchronous SGD implementations, just need to modify the worker side.
* does appear to scale slightly better from an accuracy perspective in 16-32 node counts

# cons
* Biggest criticism is the assumption of block random or round-robin worker update scheduling. Presuming each worker will update master exactly once to determine future parameter position is far from realistic on real hardware (varying capacity, performance, system loads, dealing with stragglers) and should probably be considered a synchronous not asynchronous update.
* only evaluated on image classification tasks on cifar10, cifar100, imagenet on resnet-20 and resnet-50. Would have been better to evaluate on a more varied set of tasks/models/datasets

# other comments
* Figure 2 baseline performance reported is a bit misleading/confusing since it was only evaluated on a single worker. Would suggest restricting to a single point rather than some extrapolated line that seems to indicate being run on multiple-workers.
* Figure 3 should should also show multi-node speedups for the other methods compared for completeness.
* Section 5.2 should report on percentage scaling efficiency rather than using speedup as it doesn't normalize for worker count.  For instance 16x could be interpreted as good or poor if it was achieved using 16 vs 160 nodes.
* Section 5.2 there's a small typo: GPUs -> GPU
* Consider https://arxiv.org/abs/1705.07176 in related work?

---

> ### Author Response · Authors · 2018-11-13
> **Reply to AnonReviewer3**
>
> We thank Reviewer 3 for their comments and suggestions. Here we address smaller, specific comments. Please see our other replies that address scheduling order and using more than 32 workers.
>
> 1) "Extrapolating from the curvature of the results shown it doesn't appear that DANA would continue to outperform other methods like ASGD once the worker count scales beyond the 32 node limit evaluated."
>
> This is only true for CIFAR results. CIFAR is a small dataset, and when distributed to so many workers each worker only gets very few batches. We are not aware of any distributed algorithm that achieves good accuracy with so many workers. For ImageNet, DANA scales much better than ASGD on 16 and 32 workers (Table 2), and we are now adding more workers (please see full discussion in our main reply).
>
> 2) "Section 5.2 should report on percentage scaling efficiency rather than using speedup as it doesn't normalize for worker count. For instance, 16x could be interpreted as good or poor if it was achieved using 16 vs 160 nodes."
>
> The X axis for the relevant Figure 3 shows the number of workers. We will add the necessary information to the body text.
>
> 3) "Only evaluated on image classification tasks on cifar10, cifar100, imagenet on resnet-20 and resnet-50."
>
> We did use several datasets and network architectures, but on reflection we agree with the reviewer that we may have focused too much on image classification. This was mostly because there are accepted, well-specified, easy-to-replicate benchmarks in that space. We are exploring more options, but given the time constraints and the other important experiments, we cannot guarantee to do so. We do note that nothing in DANA specifically targets any particular architecture or task.
>
> 4) We thank the reviewer for the additional reference. We note that this reference discusses convex objective functions only, while DNN training is not convex in the parameters of the network.

---

### Official Review · AnonReviewer1 · 2018-11-02
**In general, a good paper addressing an important issue with distributed deep learning training, i.e., the gradient staleness vs parallel performance (speedup).**

**Rating:** 7
**Confidence:** 4

**Review:**

The paper addresses an important problem in distributed training of deep learning models, i.e., the gradient staleness vs the parallel performance. Keeping the gradient up-to-date in distributed training is important in order to achieve a low test error and high accuracy, but that comes at a cost: the overhead of more communication and synchronization. Asynchronous methods to update the gradient have been proposed, but they usually suffer from staleness, i.e., the communication latency between the master and the slaves impacts the accuracy and training time since the accumulated gradient already is "old" in relation to the model parameters when distributed to the slaves.

The paper proposes an approach to estimate the future model parameters at the slaves using Bengio-Nesterov momentum, thus reducing the effects of the communication latency (the gap) between the master and the slaves when collecting and distributing the gradient. The novelty is mainly on the application and implementation side of the spectrum, and not so much theoretical novelty. The contribution is relatively incremental, but important and clear. Reducing training times with maintained accuracy is an important practical problem, and we need all kinds of measures to address that.

The evaluation seems solid and the results are very promising. The comparison is done with relevant "competitors" (e.g., both synchronous and asynchronous approaches for distributed training). However, since the goal of distributed learning is improved execution performance, I would have liked to see more performance numbers.

Minor:
* Page 7, top paragraph. It's written that Table 2 shows that DANA easily scales to 32 workers. That information is not shown in Table 2... You don'r show any execution time / speedup numbers at all for ImageNet input.

---

> ### Author Response · Authors · 2018-11-13
> **Reply to AnonReviewer1**
>
> We thank Reviewer 1 for their encouraging review.
>
> DANA runtime performance and speed-up are no different from standard ASGD -- DANA simply provides better final accuracy. Because ASGD performance is well-known, we did not consider performance data particularly useful for readers. We do show speedup and accuracy on a real distributed system for CIFAR-10 to drive home the point that there are no hidden surprises there (for example, the bottleneck at the master is expected and mitigations are discussed in Section 5.2).
>
> Now that we see there is an interest, we are working to provide additional speed-up and other numbers for ImageNet, and hope they will be ready in time for the deadline. Moreover, the new detailed simulation we propose (see top reply) does model runtime explicitly, and should at least allow us to show the theoretical speedups.

---

### Official Review · AnonReviewer2 · 2018-11-05
**Incremental improvement to ASGD approaches at mid-size scaling**

**Rating:** 5
**Confidence:** 3

**Review:**

Paper offers an improvement to existing approaches using momentum with SGD for asynchronous training across a distributed worker pool. The key value in the proposal seems to be that it works "out-of-the-box" and requires no new parameters to be tuned, while delivering similar final accuracy as other distributed methods.

The authors begin with an explanation of ASGD training, why it doesn't scale - worker lags that lead to gap in parameter that gradients are computed on (worker parameters) vs parameters applied (master parameters). It also discusses the kind of momentum approaches that are in use today and how it helps and hurts.

The new proposal in this paper is DANA that builds on Nesterov Momentum to reduce the lag between these two sets of parameters by predicting the parameters that should be used for computing gradients at each worker.

Pros:
- A key issue with most optimization methods is the number of hyperparameters to tune. DANA is "out-of-the-box" in that it doesn't introduce any new hyperparameters thus making it easy to scale the training of any model.

Cons:
- The sweetspot for DANA seems to be between 8-24 workers. In practice these days it is pretty easy to run synchronous SGD for these sizes with a setup of 8 GPUs per machine with a few machines. The tuning of learning rate as a hyperparameter is required anyway, and keeping training synchronous doesn't really change that. The only issue is if one often changes number of workers for training, which isn't typical.
- ASGD is useful for a larger number of workers as it is harder to train with SSGD for those because of the additional synchronization overhead. That is one area though where DANA starts to have worse behavior than other ASGD approaches.

Comments:
- Paper assumes block-random scheduling for simulation, however in practice it is quite common to have a few workers that are consistently slower. How does this kind of bias effect their methods?

---

> ### Author Response · Authors · 2018-11-13
> **Reply to AnonReviewer2**
>
> We thank Reviewer 2 for their detailed comments. Please see our other replies that address block-random scheduling order, using more than 32 workers, and the comparison to SSGD. Here we address smaller, specific comments.
>
> 1) There appears to be a misconception that a single worker must be a single GPU, and therefore it is "pretty easy to run synchronous SGD for these sizes with a setup of 8 GPUs per machine with a few machines". Putting aside the cost effectiveness of such a setup, DANA, like all ASGD algorithms, treats this as a case with 3 workers, where each worker performs SSGD internally on 8 GPUs (which is transparent to the ASGD algorithm). If 8-GPU machines are so easily available, DANA can actually train on ImageNet with (for example) 32 workers with 8 GPUs each (we again stress that the 32 worker number is not the limit of DANA, and we are adding new experiments on ImageNet to show this). We will try to make this clearer in the paper.
>
> Please see our main replies about the viability of DANA compared to SSGD, and the importance of scaling without tuning.
>
> 2) "ASGD is useful for a larger number of workers … That is one area though where DANA starts to have worse behavior than other ASGD approaches."
>
> We stress this is only true for CIFAR, a dataset too small for 32 workers: no distributed algorithm that we are aware of achieves good accuracy with so many workers. Our ImageNet results (Table 2) show that DANA is substantially better than other ASGD and SSGD approaches with 32 workers. We are now adding more workers for ImageNet to demonstrate this point.
>
> Please see full discussion in our main reply.

---

### Author Response · Authors · 2018-11-13
**1st reply to all reviewers**

We thank the anonymous reviewers for their insightful and helpful reviews. Below we address their main concerns and detail the changes we are making in response. To make discussion easier, we have grouped our replies based on common or main points raised in the reviews. We will also answer smaller, specific points in the response to each reviewer.

---

> ### Author Response · Authors · 2018-11-13
> **Reply 1.3**
>
> We respectfully disagree with reviewer #2 that "tuning of learning rate as a hyperparameter is required anyway, and keeping training synchronous doesn't really change that. The only issue is if one often changes the number of workers for training, which isn't typical.”
>
> Useful models are rarely trained once. They must occasionally be retrained down the line, for example to incorporate more recent data, or we wouldn't care about training time. The number of available workers for this retraining can change for many reasons: shared environments, limited resources, equipment malfunction, and budget increase ("we have more machines we want to use"), availability of compute servers on a public cloud, and more. Having to retune for SSGD would make this rather common case much more time consuming.
>
> Other advantages for avoiding tuning is reproducibility ("I want to be able to get the same result as someone else despite not having as many GPUS"), as well as flexibility in choosing a different infrastructure ("I want to use 20 weak GPUs instead of 4 TPUs without having to retune parameters").

---

> ### Author Response · Authors · 2018-11-13
> **Reply 1.2**
>
> Several reviewers correctly remarked that we do not show advantage beyond 32 workers.
>
> While this is true on CIFAR due to its small size (perhaps it would be more fair to say no algorithm achieves state-of-the-art accuracy with 32 workers on CIFAR), DANA can in fact converge on much more workers and to a much better accuracy than ASGD. We are now adding experiments on ImageNet with more workers -- where even at 16 workers, ASGD and SSGD already show poor convergence (Table 2). Note that even if DANA’s improvement had been limited to 16-32 workers, it is still a substantial improvement on the previous state of the art (DC-ASGD shows convergence with 8-16 workers, and we are not aware of any work that goes beyond that on CIFAR).
>
> We wish to stress DANA’s most important improvement over state-of-the-art is qualitative rather than quantitative: DANA removes the need for tuning, while other state-of-the-art algorithms rely on tuning to converge to good results. Our goal is not only to train networks faster on more workers, but to achieve good results without any tuning. Thus, the pure focus on the number of workers ("algorithm A trains on more workers than algorithm B") somewhat misses the mark. A fair quantitative comparison would need to cover the cost of tuning as well as training, but this information is seldom included in the relevant papers -- tuning is seen as either a necessity or something that one gets for free.

---

> ### Author Response · Authors · 2018-11-13
> **Reply 1.1**
>
> A common criticism was our use of block-random scheduling, which was deemed limited: it does not simulate variability in task completion time nor stragglers. We concur with the reviewers, and are now adding experiments with much more detailed simulations.
>
> The new simulations will be based on well-documented research into modeling task execution times [Ali et al., 2000]. We will repeat the experiment with the execution time for each individual batch drawn from a Gamma distribution. The Gamma distribution is a well-accepted model for task execution time, and gives rise to stragglers naturally. Our simulation will use the formulation proposed by [Ali et al., 2000] and set V=0.1 and mu=1.28  (yielding a mean execution time of 128 simulated time units, although given this is a simulation, the exact numbers do not matter).
>
> [Ali et al, 2000]: Shoukat Ali, Howard Jay Siegel, Muthucumaru Maheswaran, and Debra Hensgen. "Task execution time modeling for heterogeneous computing systems." HCW 2000.
>
> We will be happy to consider alternative approaches to simulation proposed by the reviewers.
>
> Beyond the new simulations, it is important to note that the paper does show DANA works well with real schedules. As discussed in Section 5.2, we implemented DANA on a real distributed system and its convergence in real runs in the cloud (where scheduling is natural, and we have no control over it) matched the simulated run. We agree we should do more (and are doing so), but even in the current state the reader can still have some confidence that DANA works well with realistic scheduling. Moreover, while the variability in batch completion times may affect DANA, it will of course also substantially affect ASGD and the other algorithms.

---

### Author Response · Authors · 2018-11-25
**Revision 1.0**

We thank again the anonymous reviewers for their insightful and helpful reviews which helped us improve our paper. We have updated the paper to address your concerns in the following way:

1) We added a new simulation that is based on well-documented research into modeling task execution times [Ali et al., 2000], please refer to our previous comment or the paper for details. We added Figure 3, which is similar to Figure 2 (the CIFAR experiments graphs), except that the execution time for each individual batch is drawn from a Gamma distribution. Similar to Block-Random, DANA outperforms other out-of-the-box algorithms with the new simulation.

2) We ran ImageNet on larger clusters of workers (48 and 64) and updated Table 2 accordingly. DANA outperformed all other out-of-the-box algorithms.

3) We compared the theoretical speedup of asynchronous (DANA) vs synchronous training which can be seen in Figure 5. The speedup analysis is based on the new Gamma simulation method that is mentioned in (1).

We still have some additional experiments running, and will update the paper with the results if we finish on time.

---

### Meta-Review · Area_Chair1 · 2018-12-12
**Good paper but needs more revisions**

**Confidence:** 5
**Recommendation:** Reject

**Metareview:**

The paper needs more revisions and better presentation of empirical study.